# WHEN PROMPT MEETS FREQUENCY LEARNING FOR EFFICIENT IMAGE RESTORATION

## ABSTRACT

Image restoration, as a longstanding task, aims to recover the missing details and remove degradations from a corrupted observation. Inspired by the success of prompt learning in natural language processing, many prompt-based approaches have been developed for various image restoration tasks. However, these algorithms mostly operate in the spatial domain. As frequency learning plays an important role in image restoration by reducing the spectra discrepancy between degraded/sharp image pairs, this study explores the potential of frequency prompts for efficient image restoration by proposing a plug-and-play mechanism, which mainly comprises a prompt generation module and a prompt integration module. Specifically, the former encodes different frequency information by aggregating the pre-defined learnable parameters under the guidance of implicitly decomposed spectra of input features. Subsequently, to dynamically guide reconstruction, the learned prompts are embedded into the spectra of features via dual-dimensional attention for effective frequency learning. To demonstrate the effectiveness of our mechanism, we conduct experiments on general and all-in-one image restoration tasks. By incorporating it into a CNN-based backbone, the model achieves state-of-the-art performance on **15** benchmark datasets for five representative image restoration tasks. Furthermore, equipped with our mechanism, a pure Transformer network performs favorably against state-of-the-art algorithms under two all-in-one settings.

## 1 INTRODUCTION

Due to the physical limitations of low-end sensors or terrible weather, various degradations (haze, noise, and blur) are often involved in acquired images (Su et al., 2022), which will degrade the visibility and impact the performance of models for downstream tasks. As an ill-posed problem, image restoration aims to remove those undesired degradations from observations and recover the missing details. Early approaches attempted to deal with its ill-posedness by proposing various assumptions and hand-crafted priors to reduce the solution space (Song et al., 2022; Zhang et al., 2022). In recent years, the rapid development of deep learning has spawned a great number of frameworks built on convolutional neural networks (CNNs), which can learn more robust priors from large-scale data. By means of varied advanced modules, including residual connections (Liu et al., 2019; Ruan et al., 2022), U-shaped architectures (Mao et al., 2021; Cho et al., 2021), and attention mechanisms (Qin et al., 2020; Cui et al., 2023d; Zhang et al., 2018b), these methods have achieved promising performance on multifarious image restoration tasks.

Subsequently, many general image restoration methods have been developed, which can perform well on a range of tasks after separate training (Chen et al., 2022; 2021a; Cui et al., 2023a). For example, MPRNet (Zamir et al., 2021) adopts a multi-stage CNN-based architecture to break down the recovery process into several manageable steps. Transformers have also been introduced into this track by performing self-attention within different scopes (Tsai et al., 2022; Liang et al., 2021; Wang et al., 2022; Li et al., 2023a) and dimensions (Zamir et al., 2022a). However, these solutions need individual training processes and copies when applied to different tasks, which is not practical for resource-constrained scenarios. To alleviate these issues, the recent all-in-one topic has garnered significant attention by training a unified model for a series of degradations (Lin et al., 2024; Zhang et al., 2023; Yang et al., 2023). For instance, AirNet (Li et al., 2022) is one of the pioneering methods to recover clean images in an all-in-one fashion. It works by contrastively learning the degradation representations, which are then used to restore the sharp image.

More recently, prompt learning originating from natural language processing (Zhou et al., 2022a;b) has been incorporated into general and all-in-one image restoration and has advanced performance by providing adaptive learning ability. These prompt-based methods encode the degradation priors using the produced or pre-set prompts (Potlapalli et al., 2024; Yu et al., 2024; Li et al., 2023b; Luo et al., 2021; Gao et al., 2023; Ai et al., 2023), which are then used to guide the restoration process. For example, PromptIR (Potlapalli et al., 2024) addresses all-in-one image restoration using the input-conditioned prompts that learn the knowledge of different degradation types. SelfPromer (Wang et al., 2024a) formulates the prompts based on depth cues, requiring an expensive depth estimator for supervision. However, these prompt-based methods mostly operate in the spatial domain without investigating the significance of frequency learning in prompts, which is also beneficial for high-fidelity image restoration.

In this paper, we present a plug-and-play prompt-based mechanism by formulating prompts from the perspective of frequency. To this end, we first embed crucial information about different subbands into learnable parameters with the guidance of different frequencies produced via a simple **frequency decomposition** method. Then, the learned prompts interact with the input features in the **frequency domain** via **dual-dimensional** attention weights. Overall, the resources for learning prompts and the aggregation method of injecting prompts into features are both from the frequency perspectives, resulting in consistency and effective frequency learning. Moreover, the dual-dimensional attention weights facilitate the full use of frequency signals encoded in prompts. By doing these, our mechanism enables the model to effectively recover clean images by dynamically refining frequency signals.

Our simple yet effective plug-and-play frequency prompt mechanism can be easily applied to CNN-based and Transformer-based architectures. Specifically, combined with a CNN-based backbone, our mechanism helps the model achieve state-of-the-art performance on **15** datasets for a range of representative image restoration tasks, including image dehazing, desnowing, deraining, defocus deblurring, and low-light image enhancement. Equipped with our mechanism, a pure Transformer model performs favorably against state-of-the-art algorithms under two all-in-one settings.

## 2    RELATED WORK

### 2.1    IMAGE RESTORATION

Given a corrupted image, image restoration aims to recover a clean image by removing degradations and reconstructing missing details. The development of deep learning has spawned a great number of methods, which can be roughly divided into three classes in terms of task: task-specific (Qin et al., 2020; Liu et al., 2019; Cho et al., 2021; Ruan et al., 2022), task-agnostic (or general) (Liang et al., 2021; Wang et al., 2022; Zamir et al., 2022a; Cui et al., 2023a;c), and all-in-one (Yang et al., 2023; Potlapalli et al., 2024; Yu et al., 2024; Li et al., 2023b; Luo et al., 2021). The task-specific methods can only perform well on a specific task, while task-agnostic ones can be applied to several tasks but need separate training on each dataset. These two categories have made great progress in terms of performance, which can be attributed to advanced designs for CNN- and Transformer-based frameworks. The all-in-one task has recently been a hot topic in image restoration because the all-in-one models can deal with multiple degradation types by training a single model once, which is suitable for resource-constrained scenarios. In this paper, we conduct experiments on both general and all-in-one image restoration tasks to demonstrate the effectiveness of the proposed method.

### 2.2    PROMPT LEARNING

Prompt learning is originally used in natural language processing to finetune a trained model to a downstream task by providing in-context and task-specific information. Inspired by this, this technique has been adopted in image restoration algorithms to encode degradation information (Yu et al., 2024; Li et al., 2023b; Wang et al., 2024b; Zhou et al., 2024). PromptRestorer (Wang et al., 2023) uses raw degradation features to generate prompts for general tasks and incorporates prompts from global and local perspectives via self-attention units. PromptCIR (Li et al., 2024) applies spatial prompts (Potlapalli et al., 2024) for blind compressed image restoration. SelfPromer (Wang et al., 2024a) formulates the prompt by considering the estimated depth cues for image dehazing. In the context of the all-in-one setting, PromptIR (Potlapalli et al., 2024) introduces a drop-in prompt block to dynamically adjust representations for high-fidelity image restoration. Subsequently, DA-

CLIP (Luo et al., 2021) utilizes content embedding yielded by a large vision-language model to aggregate prompts for universal image restoration. Nevertheless, these prompt-based approaches primarily leverage prompts in the spatial domain without exploring the utility in the frequency domain. In this study, to explore the potential of frequency prompts for image restoration, we use different frequencies generated via an extremely **lightweight** frequency decomposition strategy to encode frequency information into prompts, and then inject prompts into the input features in the **frequency** domain via **dual-dimensional** attention, ensuring the domain consistency between the source of producing prompts and injecting method. Furthermore, our dual-dimensional operation facilitates full use of the acquired knowledge in prompts.

## 3 FREQUENCY PROMPT MECHANISM (FPM)

Prompt learning is derived from natural language processing to achieve parameter-efficient fine-tuning of pre-trained large models to a target task. In this study, we explore the potential of this technique in the spectral domain to encode the different frequency signals for adaptive and high-fidelity image restoration. The design principle of our frequency prompt mechanism (FPM) is to generate the prompts from the frequency perspective and incorporate them into the spectra of input features for effective frequency learning. To achieve these goals, we present a prompt generation module (PGM) and a prompt integration module (PIM). Finally, FPM is realized by successively using these two modules, which can be formally expressed as:

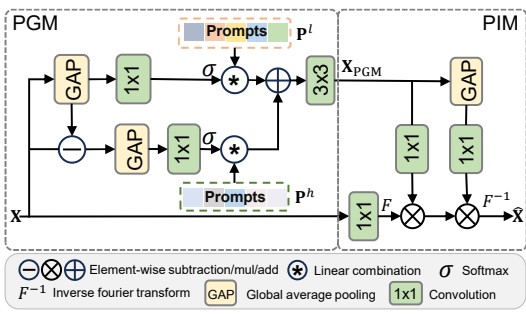

Figure 1: Illustration of our frequency prompt mechanism, containing a prompt generation module (PGM) and a prompt integration module (PIM).

$$\hat{\mathbf{X}} = \text{PIM}\left(\text{PGM}(\mathbf{X}, \mathbf{P}^l, \mathbf{P}^h), \mathbf{X}\right) \tag{1}$$

where $\mathbf{X}$ and $\hat{\mathbf{X}} \in \mathbb{R}^{C \times H \times W}$ denote the input features and output of FPM, respectively. $C, H,$ and $W$ are the channel, height, and width of features. $\mathbf{P}^l$ and $\mathbf{P}^h \in \mathbb{R}^{B \times C \times \hat{H} \times \hat{W}}$ are prompts encoding the low- and high-frequency information. $B$ specifies the number of prompts and $\hat{H} \times \hat{W}$ is resolution.

### 3.1 PROMPT GENERATION MODULE (PGM)

The PGM is responsible for encoding the informative information of different frequency subbands into prompts. To this end, we first leverage a lightweight frequency decouple method and then use the corresponding frequencies to aggregate the preset learnable parameters.

To be specific, as illustrated in Figure 1, assuming $\mathbf{X}$ is the input, a global average pooling (GAP) layer is applied to yield the low-frequency signals, where GAP serves as a kind of low-pass filter by computing the average value of the feature. Accordingly, the high-frequency component can be easily produced by removing this resulting low-frequency subband from the input $\mathbf{X}$. Subsequently, the attention weights for aggregating the prompts are produced by $1 \times 1$ convolution layers and softmax functions. After adding the prompts that are expected to encode the frequency information, the output of PGM is obtained via a $3 \times 3$ convolution. Overall, the process of PGM is formally summarized as:

$$\mathbf{X}_{\text{PGM}} = \text{Conv}_{3 \times 3}\left(\sum_{b=1}^{B} W_b^l \mathbf{P}_b^l + \sum_{b=1}^{B} W_b^h \mathbf{P}_b^h\right), \tag{2}$$

$$W^l = \text{Softmax}\left(\text{Conv}_{1 \times 1}^l\left(\text{GAP}(\mathbf{X})\right)\right), \quad W^h = \text{Softmax}\left(\text{Conv}_{1 \times 1}^h\left(\mathbf{X} - \text{GAP}(\mathbf{X})\right)\right) \tag{3}$$

where $\mathbf{X}_{\text{PGM}} \in \mathbb{R}^{C \times \hat{H} \times \hat{W}}$ is the output of PGM. GAP and Softmax are global average pooling and Softmax operators, respectively. $\text{Conv}_{1 \times 1}$ denotes a $1 \times 1$ convolution with the reduction rate of $\frac{C}{B}$ and $\text{Conv}_{3 \times 3}$ is a $3 \times 3$ convolution layer for final refinement. Before being input into PIM, $\mathbf{X}_{\text{PGM}}$ is spatially interpolated to align with the original input for further integration.

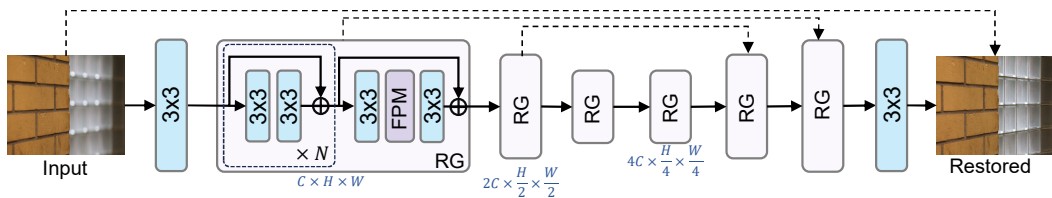

Figure 2: The CNN-based model for general image restoration. FPM is employed in a residual block.

## 3.2 PROMPT INTEGRATION MODULE (PIM)

The PIM is designed to integrate the combined frequency prompts into the original input features. Different from existing prompt-based schemes (Potlapalli et al., 2024; Wang et al., 2023; Zhou et al., 2024) that use cross-attention modules, we instead adopt a more natural solution, refining the spectra of features.

Specifically, based on the output of PGM, $\mathbf{X}_{\text{PGM}}$, PIM first uses a couple of convolution layers and GAP to generate spatial and channel attention weights, which are then utilized to modulate the Fourier spectra of the input features. This process can be formally expressed by:

$$\hat{\mathbf{X}} = F^{-1} \left( W_c W_s \left( F \left( \text{Conv}_{1\times1}(\mathbf{X}) \right) \right) \right),$$
(4)

$$W_s = \text{Conv}_{1\times1}^{\text{s}}(\mathbf{X}_{\text{PGM}}), \quad W_c = \text{Conv}_{1\times1}^{c}(\text{GAP}(\mathbf{X}_{\text{PGM}}))$$
(5)

where $F$ and $F^{-1}$ denote the fast Fourier transform and the inverse operator, respectively. $W_c \in \mathbb{R}^{C\times1\times1}$ and $W_s \in \mathbb{R}^{C\times H\times W}$ are the generated channel and spatial attention weights.

## 4 EXPERIMENTS

In this section, we conduct comprehensive experiments in two cases, *i.e.,* general and all-in-one image restoration tasks, to demonstrate the effectiveness of our proposed mechanism. In each case, we first delineate the adopted backbone and deployment method of our FPM. Next, we introduce the implementation details and used datasets. Finally, the experimental results are presented.

### 4.1 GENERAL IMAGE RESTORATION

To purely verify the efficacy of our design, we integrate our design with a classic U-shaped CNN-based backbone. As illustrated in Figure 2, the model consists of three scales. Each residual group (RG) contains $N + 1$ residual blocks, and the last one accommodates our FPM between two convolutions.

The degraded input image is processed by a $3 \times 3$ convolution layer to generate embedding features. After going through the three-scale encoder and decoder networks, the restored image is produced by another $3 \times 3$ convolution layer and image-level residual connection.

#### 4.1.1 IMPLEMENTATION DETAILS

For general image restoration, we conduct experiments by separately training models on different datasets for five representative image restoration tasks. Specifically, for dehazing, we evaluate our methods on synthetic daytime datasets (SOTS-Indoor (Li et al., 2018), SOTS-Outdoor (Li et al., 2018), Haze4K (Liu et al., 2021)), nighttime datasets (GTA5 (Yan et al., 2020), NHR (Zhang et al., 2020)), remote sensing datasets (SateHaze1k (Huang et al., 2020)-Thin, Moderate, Thick), and a real-world dataset (DenseHaze (Ancuti et al., 2019)). Moreover, the models are tested on the widely used CSD (Chen et al., 2021b), SRRS (Chen et al., 2020), and Snow100K (Liu et al., 2018) for image desnowing, DPDD (Abuolaim & Brown, 2020) for defocus deblurring, LOL-v2-Synthetic (Yang et al., 2021) for low-light image enhancement, and Test2800 (Fu et al., 2017) for deraining.

The model is trained using the Adam (Kingma & Ba, 2014) optimizer and dual-domain $L_1$ loss functions (Cho et al., 2021; Cui et al., 2023a). Random horizontal flips are used for data augmentation. Following previous methods (Cui et al., 2023a; Zamir et al., 2021), we use different numbers of

Table 1: Image dehazing comparisons on the synthetic daytime datasets (SOTS-Indoor (Li et al., 2018), SOTS-Outdoor (Li et al., 2018)) and a real-world dataset (Dense-Haze (Ancuti et al., 2019)).

| Method | SOTS-Indoor | | SOTS-Outdoor | | Dense-Haze | | Average | |
|---|---|---|---|---|---|---|---|---|
| | PSNR | SSIM | PSNR | SSIM | PSNR | SSIM | PSNR | SSIM |
| MSBDN (Dong et al., 2020a) | 33.67 | 0.985 | 33.48 | 0.982 | 15.37 | 0.49 | 27.51 | 0.819 |
| FFA-Net (Qin et al., 2020) | 36.39 | 0.989 | 33.57 | 0.984 | 14.39 | 0.45 | 28.12 | 0.808 |
| AECR-Net (Wu et al., 2021) | 37.17 | 0.990 | - | - | 15.80 | 0.47 | - | - |
| DeHamer (Guo et al., 2022) | 36.63 | 0.988 | 35.18 | 0.986 | 16.62 | 0.56 | 29.48 | 0.845 |
| PMNet (Ye et al., 2022) | 38.41 | 0.990 | 34.74 | 0.985 | 16.79 | 0.51 | 29.98 | 0.828 |
| MAXIM (Tu et al., 2022) | 38.11 | 0.991 | 34.19 | 0.985 | - | - | - | - |
| FocalNet (Cui et al., 2023a) | 40.82 | 0.996 | 37.71 | 0.995 | 17.07 | 0.63 | 31.87 | 0.874 |
| DEA-Net (Chen et al., 2024) | 40.20 | 0.993 | 36.03 | 0.989 | - | - | - | - |
| FSNet-S (Cui et al., 2023b) | 40.47 | 0.996 | 37.24 | 0.994 | 17.00 | 0.65 | 31.57 | 0.880 |
| MB-TaylorFormer-B (Qiu et al., 2023) | 40.71 | 0.992 | 37.42 | 0.989 | 16.66 | 0.56 | 31.60 | 0.847 |
| **Ours** | **40.86** | **0.996** | **37.86** | **0.995** | **17.33** | **0.65** | **32.02** | **0.880** |

Table 2: Image dehazing comparisons on the remote sensing SateHaze1k (Huang et al., 2020) dataset. The models are separately trained and tested on each subset.

| Methods | Thin | | Moderate | | Thick | | Average | |
|---|---|---|---|---|---|---|---|---|
| | PSNR | SSIM | PSNR | SSIM | PSNR | SSIM | PSNR | SSIM |
| AOD-Net (Li et al., 2017) | 19.54 | 0.854 | 20.10 | 0.885 | 15.92 | 0.731 | 18.52 | 0.823 |
| H2RL-Net (Chen et al., 2021c) | 20.91 | 0.880 | 22.34 | 0.906 | 17.41 | 0.768 | 20.22 | 0.851 |
| FCFT-Net (Li & Chen, 2020) | 23.59 | 0.913 | 22.88 | 0.927 | 20.03 | 0.816 | 22.17 | 0.885 |
| Uformer (Wang et al., 2022) | 22.82 | 0.907 | 24.47 | 0.939 | 20.36 | 0.815 | 22.55 | 0.887 |
| $C^2$PNet (Zheng et al., 2023) | 19.62 | 0.880 | 24.79 | 0.940 | 16.83 | 0.790 | 20.41 | 0.870 |
| Restormer (Zamir et al., 2022a) | 23.08 | 0.912 | 24.73 | 0.933 | 18.58 | 0.762 | 22.13 | 0.869 |
| Trinity-Net (Chi et al., 2023) | 21.55 | 0.884 | 23.35 | 0.895 | 20.97 | 0.823 | 21.96 | 0.867 |
| FocalNet (Cui et al., 2023a) | 24.16 | 0.916 | 25.99 | 0.947 | 21.69 | 0.847 | 23.95 | 0.903 |
| **Ours** | **24.27** | **0.976** | **26.42** | **0.978** | **22.81** | **0.955** | **24.50** | **0.970** |

Table 3: Image dehazing comparisons on the Haze4K (Liu et al., 2021) dataset.

| Method | DehazeNet | AOD-Net | GDN | MSBDN | FFA-Net | PMNet | FSNet | **Ours** |
|---|---|---|---|---|---|---|---|---|
| PSNR | 19.12 | 17.15 | 23.29 | 22.99 | 26.96 | 33.49 | 34.12 | **34.14** |
| SSIM | 0.84 | 0.83 | 0.93 | 0.85 | 0.95 | 0.98 | 0.99 | **0.99** |

Table 4: Image dehazing comparisons on the nighttime NHR (Zhang et al., 2020) dataset.

| Method | GS | MRPF | MRP | OSFD | HCD | FSNet-S | FocalNet | **Ours** |
|---|---|---|---|---|---|---|---|---|
| PSNR | 17.32 | 16.95 | 19.93 | 21.32 | 23.43 | 24.35 | 25.35 | **26.24** |
| SSIM | 0.629 | 0.667 | 0.777 | 0.804 | 0.953 | 0.965 | 0.969 | **0.972** |

Table 5: Image dehazing comparisons on the nighttime GTA5 (Yan et al., 2020) dataset.

| Method | GS | MRP | Ancuti *et al.* | Yan *et al.* | CycleGAN | Jin *et al.* | FocalNet | **Ours** |
|---|---|---|---|---|---|---|---|---|
| PSNR | 21.02 | 20.92 | 20.59 | 27.00 | 21.75 | 30.38 | 30.65 | **30.73** |
| SSIM | 0.639 | 0.646 | 0.623 | 0.850 | 0.696 | 0.904 | 0.909 | **0.911** |

residual blocks in each RG according to the complexity of tasks. The number of prompts $B$ is set to 5. All experiments are performed on an NVIDIA Tesla A100 GPU. More details of the used datasets and specific training configurations are provided in the Appendix.

Table 6: Image desnowing comparisons on CSD (Chen et al., 2021b), SRRS (Chen et al., 2020), and Snow100K (Liu et al., 2018).

| Method | CSD | | SRRS | | Snow100K | | Average | |
|---|---|---|---|---|---|---|---|---|
| | PSNR | SSIM | PSNR | SSIM | PSNR | SSIM | PSNR | SSIM |
| DesnowNet (Liu et al., 2018) | 20.13 | 0.81 | 20.38 | 0.84 | 30.50 | 0.94 | 23.67 | 0.86 |
| All in One (Li et al., 2020) | 26.31 | 0.87 | 24.98 | 0.88 | 26.07 | 0.88 | 25.79 | 0.88 |
| JSTASR (Chen et al., 2020) | 27.96 | 0.88 | 25.82 | 0.89 | 23.12 | 0.86 | 25.63 | 0.88 |
| HDCW-Net (Chen et al., 2021b) | 29.06 | 0.91 | 27.78 | 0.92 | 31.54 | 0.95 | 29.46 | 0.93 |
| MSP-Former (Chen et al., 2023) | 33.75 | 0.96 | 30.76 | 0.95 | 33.43 | **0.96** | 32.65 | 0.96 |
| TransWeather (Valanarasu et al., 2022) | 31.76 | 0.93 | 28.29 | 0.92 | 31.82 | 0.93 | 30.62 | 0.93 |
| FSNet-S (Cui et al., 2023b) | 35.33 | 0.98 | 31.39 | 0.98 | 33.36 | 0.95 | 33.36 | 0.97 |
| FocalNet (Cui et al., 2023a) | 37.18 | 0.99 | 31.34 | 0.98 | 33.53 | 0.95 | 34.02 | 0.97 |
| **Ours** | **37.31** | **0.99** | **31.78** | **0.98** | **33.61** | 0.95 | **34.23** | **0.97** |

Table 7: Image defocus deblurring comparisons on the DPDD (Abuolaim & Brown, 2020) dataset.

| Method | Indoor Scenes | | | | Outdoor Scenes | | | | Combined | | | |
|---|---|---|---|---|---|---|---|---|---|---|---|---|
| | PSNR↑ | SSIM↑ | MAE↓ | LPIPS↓ | PSNR | SSIM | MAE | LPIPS | PSNR | SSIM | MAE | LPIPS |
| EBDB (Karaali & Jung, 2017) | 25.77 | 0.772 | 0.040 | 0.297 | 21.25 | 0.599 | 0.058 | 0.373 | 23.45 | 0.683 | 0.049 | 0.336 |
| DMENet (Lee et al., 2019) | 25.50 | 0.788 | 0.038 | 0.298 | 21.43 | 0.644 | 0.063 | 0.397 | 23.41 | 0.714 | 0.051 | 0.349 |
| JNB (Shi et al., 2015) | 26.73 | 0.828 | 0.031 | 0.273 | 21.10 | 0.608 | 0.064 | 0.355 | 23.84 | 0.715 | 0.048 | 0.315 |
| DPDNet (Abuolaim & Brown, 2020) | 26.54 | 0.816 | 0.031 | 0.239 | 22.25 | 0.682 | 0.056 | 0.313 | 24.34 | 0.747 | 0.044 | 0.277 |
| KPAC (Son et al., 2021) | 27.97 | 0.852 | 0.026 | 0.182 | 22.62 | 0.701 | 0.053 | 0.269 | 25.22 | 0.774 | 0.040 | 0.227 |
| IFAN (Lee et al., 2021) | 28.11 | 0.861 | 0.026 | 0.179 | 22.76 | 0.720 | 0.052 | 0.254 | 25.37 | 0.789 | 0.039 | 0.217 |
| DRBNet (Ruan et al., 2022) | - | | | | - | | | | 25.73 | 0.791 | - | 0.183 |
| Restormer (Zamir et al., 2022a) | 28.87 | 0.882 | 0.025 | 0.145 | 23.24 | 0.743 | 0.050 | 0.209 | 25.98 | 0.811 | 0.038 | 0.178 |
| FocalNet (Cui et al., 2023a) | 29.10 | 0.876 | 0.024 | 0.173 | 23.41 | 0.743 | 0.049 | 0.246 | 26.18 | 0.808 | 0.037 | 0.210 |
| Lin *et al* (Lin et al., 2024) | 29.11 | **0.889** | - | - | 23.35 | 0.748 | - | - | 26.15 | **0.817** | - | - |
| FSNet (Cui et al., 2023b) | 29.14 | 0.878 | 0.024 | 0.166 | 23.45 | 0.747 | 0.050 | 0.246 | 26.22 | 0.811 | 0.037 | 0.207 |
| **Ours** | **29.38** | 0.883 | **0.023** | **0.145** | **23.49** | **0.753** | 0.049 | **0.208** | **26.35** | 0.816 | **0.036** | **0.178** |

Table 8: Low-light image enhancement results on the LOL-v2 (Yang et al., 2021) dataset.

| Method | EnGAN | RUAS | FIDE | DRBN | KinD | Restormer | MIRNet | SNR-Net | Retinexformer | Ours |
|---|---|---|---|---|---|---|---|---|---|---|
| PSNR | 16.57 | 16.55 | 15.20 | 23.22 | 13.29 | 21.41 | 21.94 | 24.14 | 25.67 | **26.21** |
| SSIM | 0.734 | 0.652 | 0.612 | 0.927 | 0.578 | 0.830 | 0.876 | 0.928 | 0.930 | **0.958** |

Table 9: Image deraining comparisons on the Test2800 (Fu et al., 2017) dataset.

| Method | DerainNet | UMRL | RESCAN | PreNet | MSPFN | MPRNet | FSNet | **Our** |
|---|---|---|---|---|---|---|---|---|
| PSNR | 24.31 | 29.97 | 31.29 | 31.75 | 32.82 | 33.64 | 33.64 | **33.72** |
| SSIM | 0.861 | 0.905 | 0.904 | 0.916 | 0.930 | 0.938 | 0.936 | **0.937** |

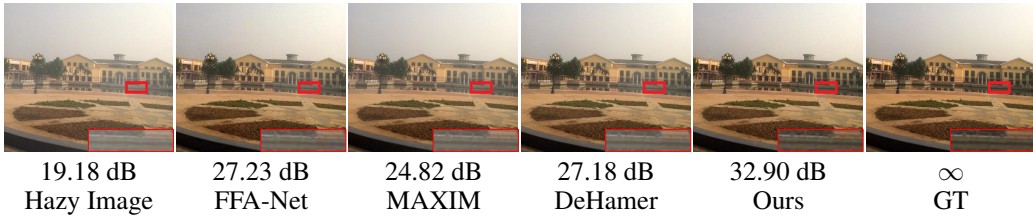

| 19.18 dB | 27.23 dB | 24.82 dB | 27.18 dB | 32.90 dB | ∞ |
|---|---|---|---|---|---|
| Hazy Image | FFA-Net | MAXIM | DeHamer | Ours | GT |

Figure 3: Image dehazing comparisons on the SOTS-Outdoor (Li et al., 2018) dataset.

### 4.1.2 EXPERIMENTAL RESULTS

The quantitative results for image dehazing, desnowing, defocus deblurring, low-light image enhancement, and deraining are presented in Table 1-5, Table 6, Table 7, Table 8, and Table 9, respectively. The best scores in the tables are highlighted in **bold**. From the tables, we can see that our network achieves the best performance on most metrics. It is worth mentioning that our results are obtained

Table 10: Break-down ablations.

| Method | Baseline | PGM | PIM | Full |
|--------|----------|-----|-----|------|
| PSNR | 31.33 | 33.82 | 34.54 | 35.18 |
| GFLOPs | 15.44 | 19.06 | 16.26 | 19.89 |

Table 11: Different alternatives to PIM.

| Method | Spatial Domain | Cross Attention | PIM w/o Channel | Ours |
|--------|----------------|-----------------|-----------------|------|
| PSNR | 34.25 | 34.17 | 34.85 | **35.18** |

Table 12: Frequencies used to aggregate prompts in PGM.

| Net | Frequency | PSNR |
|-----|-----------|------|
| (a) | Baseline | 31.33 |
| (b) | Low | 33.24 |
| (c) | High | 31.47 |
| (d) | None | 32.34 |
| (e) | Low/Low | 33.72 |
| (f) | High/High | 33.16 |
| (g) | Ours | **33.82** |

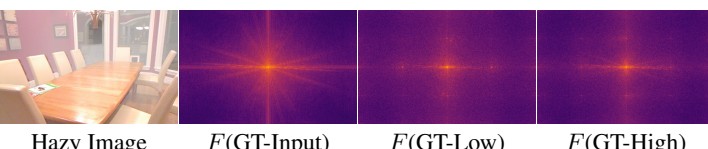

Hazy Image    $F$(GT-Input)    $F$(GT-Low)    $F$(GT-High)

Figure 4: The spectra of differences between ground truth and hazy image, low-frequency prompt and high-frequency prompt results. The latter two images are produced using only the low-frequency and high-frequency prompts, respectively, during the inference of our dehazing model. $F$ is the fast Fourier transform. Our frequency prompts can recover the corresponding frequency signals.

by merging our proposed mechanism into a simple CNN-based backbone. Figure 3 shows that our model generates a more favorable image than competitors by removing more haze degradations. These results demonstrate that our novel design is beneficial for high-fidelity image restoration. Due to the space limit, the qualitative comparisons for more tasks are presented in the Appendix.

### 4.1.3 ABLATION STUDIES

We perform the ablation results by training a dehazing model ($N = 0$) on RESIDE-Indoor (Li et al., 2018) for 300 epochs and testing on SOTS-Indoor (Li et al., 2018). More ablation studies can be found in the Appendix.

**Effects of individual components.** The baseline model is obtained by removing our FPM from the dehazing model. Table 10 shows that the baseline model achieves 31.33 dB PSNR on the SOTS-Indoor (Li et al., 2018) dataset. Next, equipped with PGM, the model achieves a gain of 2.49 dB PSNR by directly adding the output of PGM to input features in the spatial domain. The PIM version, taking the original input features as input and imposing the attention weights on input features in the spectral domain, outperforms the baseline by 3.21 dB PSNR. Taken together, the full model obtains the best performance, suggesting the effectiveness of frequency-inspired prompt design.

**Design choices for integration method in PIM.** We further explore the influence of the integration method in PIM by performing experiments with several alternatives. Table 11 shows that applying the attention weights produced by PIM to input features in the spatial domain results in a degradation of 0.93 dB PSNR compared to our frequency version. The widely adopted cross-attention method in existing prompt-based algorithms only achieves 34.17 dB PSNR. Using only the spatial attention in PIM is superior to our bi-dimensional variant. The results reveal that, besides the generation of prompts, the integration method also plays a significant role in exploring the potential of prompts.

**Design choices for PGM.** To verify the efficacy of our PGM design, we experiment using different combinations of frequencies to aggregate prompts. Table 12 shows that the model using the low-frequency or high-frequency prompts both outperforms the baseline model, demonstrating the effectiveness of prompt learning for image restoration. The input-conditioned prompts (Table 12 e,f) are superior to the counterpart, *None* (Table 12 d), where the preset learnable parameters are directly injected into the input features without the guidance of frequency-based attention weights. Our design, employing different frequencies for guidance, attains the best performance.

**Visual results of our mechanism.** To understand the mechanisms of our frequency prompt design more intuitively, we compute the Fourier spectra of differences between ground truth and the input image, low-frequency and high-frequency prompt results. For example, the low-frequency result is obtained by removing the high-frequency prompts during the inference of our dehazing model.

Figure 5: The Transformer model used for all-in-one image restoration. FPM is deployed in the decoder stage following (Potlapalli et al., 2024). The architecture of the Transformer Block (TB) is consistent with (Zamir et al., 2022a).

Table 13: The datasets for three-task and five-task settings. Entries with † are exclusively used for the five-task setting. The noisy images are yielded by adding Gaussian noise of level $\sigma \in \{15, 25, 50\}$.

| Task | Denoising | Deraining | Dehazing | Deblurring† | Low-light† |
|------|-----------|-----------|----------|-------------|-----------|
| Train | BSD400, WED | Rain100L | RESIDE-$\beta$ | GoPro | LOL-v1 |
| Test | BSD68,Urban100,Kodak24† | Rain100L | SOTS-Outdoor | GoPro | LOL-v1 |

Table 14: Comparisons under the three-task setting. A unified model is trained on compound datasets.

| Method | Denoising on BSD68 | | | | | | Deraining on Rain100L | | Dehazing on SOTS | | Average | |
| | $\sigma = 15$ | | $\sigma = 25$ | | $\sigma = 50$ | | | | | | | |
| | PSNR | SSIM | PSNR | SSIM | PSNR | SSIM | PSNR | SSIM | PSNR | SSIM | PSNR | SSIM |
|--------|------|------|------|------|------|------|------|------|------|------|------|------|
| BRDNet (Tian et al., 2020) | 32.26 | 0.898 | 29.76 | 0.836 | 26.34 | 0.693 | 27.42 | 0.895 | 23.23 | 0.895 | 27.80 | 0.843 |
| LPNet (Gao et al., 2019) | 26.47 | 0.778 | 24.77 | 0.748 | 21.26 | 0.552 | 24.88 | 0.784 | 20.84 | 0.828 | 23.64 | 0.738 |
| FDGAN (Dong et al., 2020b) | 30.25 | 0.910 | 28.81 | 0.868 | 26.43 | 0.776 | 29.89 | 0.933 | 24.71 | 0.929 | 28.02 | 0.883 |
| MPRNet (Zamir et al., 2021) | 33.54 | 0.927 | 30.89 | 0.880 | 27.56 | 0.779 | 33.57 | 0.954 | 25.28 | 0.955 | 30.17 | 0.899 |
| DL (Fan et al., 2019) | 33.05 | 0.914 | 30.41 | 0.861 | 26.90 | 0.740 | 32.62 | 0.931 | 26.92 | 0.931 | 29.98 | 0.876 |
| AirNet (Li et al., 2022) | 33.92 | 0.933 | 31.26 | 0.888 | 28.00 | 0.797 | 34.90 | 0.968 | 27.94 | 0.962 | 31.20 | 0.910 |
| PromptIR (Potlapalli et al., 2024) | 33.98 | 0.933 | 31.31 | 0.888 | 28.06 | 0.799 | 36.37 | 0.972 | 30.58 | 0.974 | 32.06 | 0.913 |
| Lin *et al* (Lin et al., 2024) | 34.01 | 0.933 | 31.39 | 0.890 | 28.18 | 0.802 | 37.58 | 0.979 | **31.63** | **0.980** | 32.56 | 0.916 |
| **Ours** | **34.11** | **0.935** | **31.45** | **0.891** | **28.19** | **0.802** | **38.58** | **0.982** | 30.85 | 0.979 | **32.64** | **0.918** |

Figure 4 shows that frequency prompts effectively recover the corresponding frequency information. For example, the difference in the high-frequency segments is reduced by our high-frequency prompts.

## 4.2 ALL-IN-ONE IMAGE RESTORATION

We further integrate our FPM into a plain Transformer-based backbone (Zamir et al., 2022a) to demonstrate the effectiveness of our method in all-in-one image restoration, which also suggests the adaptability of our FPM to different architectures. Figure 5 showcases the encoder-decoder architecture of the used Transformer block, where FPM is employed only in the decoder stage, following (Potlapalli et al., 2024).

### 4.2.1 IMPLEMENTATION DETAILS

We evaluate the established model in the three-task setting (Li et al., 2024) (denoising, deraining, dehazing) and five-task setting (Zhang et al., 2023), where motion deblurring and low-light image enhancement are additionally adopted. To train a single model under the three-task or five-task setting, we combine the datasets of those tasks for training (see Table 13). The model is then evaluated using the corresponding test sets of each task. Following (Li et al., 2024), we also experiment under a single-task setting, where the model is individually trained and evaluated for each task.

The all-in-one model is trained using Adam (Kingma & Ba, 2014) with a batch size of 32 for 150 epochs. The learning rate is $2e^{-4}$ and the patch size is $3 \times 128 \times 128$. Random horizontal and vertical flips are adopted for data augmentation. $L_1$ to $L_5$ in the model are set to 4, 6, 6, 8, and 8, respectively. Regarding the single-task setting, training setups remain unchanged except for the batch size, which is set to 8.

Table 15: Image deraining comparisons on Rain100L (Yang et al., 2019) for the single-task setting.

| Method | UMR | SIRR | MSPFN | LPNet | AirNet | Restormer | PromptIR | **Ours** |
|--------|-----|------|-------|-------|--------|-----------|----------|----------|
| PSNR | 32.39 | 32.37 | 33.50 | 33.61 | 34.90 | 36.74 | 37.04 | **39.03** |
| SSIM | 0.921 | 0.926 | 0.948 | 0.958 | 0.977 | 0.978 | 0.979 | **0.985** |

Table 16: Image dehazing comparisons in the single-task setting on SOTS-Outdoor (Li et al., 2018).

| Method | DehazeNet | AODNet | EPDN | FDGAN | AirNet | Restormer | PromptIR | **Ours** |
|--------|-----------|--------|------|-------|--------|-----------|----------|----------|
| PSNR | 22.46 | 20.29 | 22.57 | 23.15 | 23.18 | 30.87 | 31.31 | **31.66** |
| SSIM | 0.851 | 0.877 | 0.863 | 0.921 | 0.900 | 0.969 | 0.973 | **0.981** |

Table 17: Denoising scores (PSNR/SSIM) for Urban100 (Huang et al., 2015) and BSD68 (Martin et al., 2001) in single-task setting.

| Method | Urban100 | | | BSD68 | | | |
|--------|----------|----------|----------|-------|-------|-------|---------|
| | $\sigma = 15$ | $\sigma = 25$ | $\sigma = 50$ | $\sigma = 15$ | $\sigma = 25$ | $\sigma = 50$ | Average |
| CBM3D (Dabov et al., 2007) | 33.93/0.941 | 31.36/0.909 | 27.93/0.840 | 33.50/0.922 | 30.69/0.868 | 27.36/0.763 | 30.80/0.874 |
| DnCNN (Zhang et al., 2017a) | 32.98/0.931 | 30.81/0.902 | 27.59/0.833 | 33.89/0.930 | 31.23/0.883 | 27.92/0.789 | 30.74/0.878 |
| IRCNN (Zhang et al., 2017b) | 27.59/0.833 | 31.20/0.909 | 27.70/0.840 | 33.87/0.929 | 31.18/0.882 | 27.88/0.790 | 29.90/0.864 |
| FFDNet (Zhang et al., 2018a) | 33.83/0.942 | 31.40/0.912 | 28.05/0.848 | 33.87/0.929 | 31.21/0.882 | 27.96/0.789 | 31.05/0.884 |
| BRDNet (Tian et al., 2020) | 34.42/0.946 | 31.99/0.919 | 28.56/0.858 | 34.10/0.929 | 31.43/0.885 | 28.16/0.794 | 31.44/0.889 |
| AirNet (Li et al., 2022) | 34.40/0.949 | 32.10/0.924 | 28.88/0.871 | 34.14/0.936 | 31.48/0.893 | 28.23/0.806 | 31.54/0.897 |
| PromptIR (Potlapalli et al., 2024) | 34.77/0.952 | 32.49/0.929 | 29.39/0.881 | 34.34/0.938 | 31.71/0.897 | 28.49/0.813 | 31.87/0.902 |
| **Ours** | **34.91/0.952** | **32.74/0.931** | **29.72/0.886** | **34.35/0.938** | **31.71/0.897** | **28.50/0.814** | **31.99/0.903** |

Table 18: Results (PSNR/SSIM) for the five-task setting. Denoising scores are computed with $\sigma = 25$. The first and second super-lists include the generate and all-in-one restoration methods, respectively.

| Method | Dehazing on SOTS | Deraining on Rain100L | Denoising on BSD68 | Deblurring on GoPro | Low-Light on LOL | Average |
|--------|------------------|-----------------------|--------------------|---------------------|------------------|---------|
| NAFNet (Chen et al., 2022) | 25.23/0.939 | 35.56/0.967 | 31.02/0.883 | 26.53/0.808 | 20.49/0.809 | 27.76/0.881 |
| HINet (Chen et al., 2021a) | 24.74/0.937 | 35.67/0.969 | 31.00/0.881 | 26.12/0.788 | 19.47/0.800 | 27.40/0.875 |
| MPRNet (Zamir et al., 2021) | 24.27/0.937 | **38.16/0.981** | 31.35/**0.889** | 26.87/0.823 | 20.84/0.824 | 28.27/0.890 |
| DGUNet (Mou et al., 2022) | 24.78/0.940 | 36.62/0.971 | 31.10/0.883 | 27.25/0.837 | 21.87/0.823 | 28.32/0.891 |
| MIRNetV2 (Zamir et al., 2022b) | 24.03/0.927 | 33.89/0.954 | 30.97/0.881 | 26.30/0.799 | 21.52/0.815 | 27.34/0.875 |
| SwinIR (Liang et al., 2021) | 21.50/0.891 | 30.78/0.923 | 30.59/0.868 | 24.52/0.773 | 17.81/0.723 | 25.04/0.835 |
| Restormer (Zamir et al., 2022a) | 24.09/0.927 | 34.81/0.962 | 31.49/0.884 | 27.22/0.829 | 20.41/0.806 | 27.60/0.881 |
| DL (Fan et al., 2019) | 20.54/0.826 | 21.96/0.762 | 23.09/0.745 | 19.86/0.672 | 19.83/0.712 | 21.05/0.743 |
| Transweather (Valanarasu et al., 2022) | 21.32/0.885 | 29.43/0.905 | 29.00/0.841 | 25.12/0.757 | 21.21/0.792 | 25.22/0.836 |
| TAPE (Liu et al., 2022) | 22.16/0.861 | 29.67/0.904 | 30.18/0.855 | 24.47/0.763 | 18.97/0.621 | 25.09/0.801 |
| AirNet (Li et al., 2022) | 21.04/0.884 | 32.98/0.951 | 30.91/0.882 | 24.35/0.781 | 18.18/0.735 | 25.49/0.846 |
| IDR (Zhang et al., 2023) | 25.24/0.943 | 35.63/0.965 | **31.60**/0.887 | 27.87/0.846 | 21.34/0.826 | 28.34/0.893 |
| **Ours** | **30.90/0.980** | 37.99/**0.981** | 31.34/0.888 | **28.32/0.862** | **23.21/0.846** | **30.35/0.911** |

### 4.2.2 EXPERIMENTAL RESULTS

The quantitative results for the three-task setting are presented in Table 14. Our model performs well on most datasets and metrics. Particularly on the Rain100L (Yang et al., 2019) dataset for image deraining, the performance gain can be as significant as 1 dB PSNR compared to the second-best method (Lin et al., 2024). Moreover, following previous schemes (Li et al., 2022; 2024), we evaluate our model under the single-task setting by training models individually for each task. The results for image deraining, dehazing, and denoising are reported in Table 15, Table 16, and Table 17, respectively. Our model achieves 2.99 dB, 0.35 dB, and 0.12 dB performance gains over the recent state-of-the-art PromptIR (Li et al., 2024) algorithm, which employs spatial prompts. These results demonstrate the superiority of our design.

In addition, we provide the comparisons under the five-task setting in Table 18. Our method is superior to competitors on most tasks. Specifically, our method achieves an average performance

Table 19: Image denoising results (PSNR) on Urban100 (Huang et al., 2015) and Kodak24 (Franzen, 1999). The scores are obtained by directly using the model trained under the five-task setting.

| Method | Urban100 | | | Kodak24 | | | |
|---|---|---|---|---|---|---|---|
| | $\sigma = 15$ | $\sigma = 25$ | $\sigma = 50$ | $\sigma = 15$ | $\sigma = 25$ | $\sigma = 50$ | Average |
| DL (Fan et al., 2019) | 21.10 | 21.28 | 20.42 | 22.63 | 22.66 | 21.95 | 21.67 |
| Transweather (Valanarasu et al., 2022) | 29.64 | 27.97 | 26.08 | 31.67 | 29.64 | 26.74 | 28.62 |
| TAPE (Liu et al., 2022) | 32.19 | 29.65 | 25.87 | 33.24 | 30.70 | 27.19 | 29.81 |
| AirNet (Li et al., 2022) | 33.16 | 30.83 | 27.45 | 34.14 | 31.74 | 28.59 | 30.99 |
| IDR (Zhang et al., 2023) | 33.82 | 31.29 | 28.07 | 34.78 | **32.42** | 29.13 | 31.59 |
| **Ours** | **34.08** | **31.67** | **28.29** | **34.89** | 32.39 | **29.22** | **31.76** |

Table 20: Ablation studies under the all-in-one setting.

| Method | Dehazing on SOTS | | Deraining on Rain100L | | Denoising on BSD68 | | | | | | Average | |
| | | | | | $\sigma = 15$ | | $\sigma = 25$ | | $\sigma = 50$ | | | |
| | PSNR | SSIM | PSNR | SSIM | PSNR | SSIM | PSNR | SSIM | PSNR | SSIM | PSNR | SSIM |
|---|---|---|---|---|---|---|---|---|---|---|---|---|
| Baseline | 28.37 | 0.956 | 35.28 | 0.964 | 33.82 | 0.930 | 31.16 | 0.884 | 27.84 | 0.785 | 31.29 | 0.904 |
| PGM | 30.17 | 0.974 | 36.67 | 0.973 | 33.82 | 0.931 | 31.16 | 0.884 | 27.88 | 0.788 | 31.94 | 0.910 |
| PIM | 30.15 | 0.977 | 37.13 | 0.977 | 33.85 | 0.931 | 31.18 | 0.885 | 27.91 | 0.791 | 32.04 | 0.912 |
| Full (FPM) | 30.97 | 0.978 | 37.16 | 0.978 | 33.88 | 0.932 | 31.21 | 0.887 | 27.94 | 0.792 | 32.23 | 0.913 |

gain of 2.01 dB PSNR over IDR (Zhang et al., 2023). Especially for dehazing on SOTS (Li et al., 2018), the advantage can reach 5.66 dB PSNR. These results indicate the effectiveness of our method. We provide visualizations for all-in-one settings in the Appendix.

Furthermore, we directly apply the model trained under the five-task setting to two out-of-distribution denoising datasets. Table 19 shows that our model has a stronger generalization ability than IDR (Zhang et al., 2023) by producing an average gain of 0.17 dB PSNR. Especially on the Urban100 (Huang et al., 2015) dataset, the advantage can be as large as 0.38 dB PSNR for $\sigma = 25$.

### 4.2.3 ABLATION STUDIES

We perform ablation studies by training the model for 30 epochs under the three-task setting to demonstrate the effectiveness of our FPM in all-in-one settings. Table 20 shows the Transformer baseline model achieves an average PSNR of 31.29 dB. Our PIM improves performance on all degradation types and levels, resulting in an average gain of 0.75 dB in PSNR. The complete model, incorporating PIM and PGM, further boosts the performance on all datasets, suggesting the efficacy of our design.

## 5 CONCLUSION

This paper presents a frequency prompt mechanism (FPM) for image restoration, which is designed by using the implicitly decomposed frequency signals to guide the aggregation of predefined learnable parameters and injecting the learned prompts into the spectra of original input features. More concretely, the prompt generation module leverages the global average pooling to decouple features into different frequency segments for guidance to make prompts encode informative low- and high-frequency information. The obtained prompts are then incorporated with input features via bi-dimensional attention in the spectral domain. FPM can be employed in different architectures. Built on it, the CNN-based network achieves state-of-the-art performance on 15 datasets for five representative image restoration tasks, including image dehazing, defocus deblurring, desnowing, deraining, and low-light image enhancement. Furthermore, equipped with FPM, the Transformer backbone performs favorably against state-of-the-art algorithms in all-in-one restoration settings. These results demonstrate that our design is a valuable contribution to the realm of image restoration.

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

APPENDIX

This appendix provides specific training configurations for general image restoration, more ablation studies, computational comparisons, and visual comparisons.

## A    DATASETS AND TRAINING CONFIGURATIONS

In this section, we provide more details of the used datasets and specific training configurations for different general image restoration tasks. According to the complexity of different tasks, we set $N$ to 3 for tasks of dehazing, desnowing, and low-light image enhancement and 15 for deblurring and deraining. Unless specified otherwise, the patch size and batch size adopted for training are $3 \times 256 \times 256$ and 8, respectively. The initial learning rate is set to $2e^{-4}$, which is reduced to $1e^{-6}$ with the cosine annealing strategy.

**Image Dehazing.** We evaluate our method on four kinds of datasets: synthetic daytime datasets, a real-world dataset, nighttime datasets, and remote sensing datasets. For daytime scenes, we use the widely adopted RESIDE-Indoor (Li et al., 2018) and RESIDE-Outdoor (Li et al., 2018) datasets for training and evaluation. Specifically, the model is trained on these two datasets for 1000 epochs and 30 epochs, respectively, and then tested on the corresponding test sets: SOTS-Indoor (Li et al., 2018) and SOTS-Outdoor (Li et al., 2018). The initial learning rate for RESIDE-Outdoor is set to $1e^{-4}$. Moreover, our method is trained and evaluated on a more realistic synthetic dataset, Haze4K (Liu et al., 2021). The model is trained for 1000 epochs with a batch size of 4 and a learning rate of $1e^{-4}$. Furthermore, our model is trained and tested on the Dense-Haze (Ancuti et al., 2019) datasets to evaluate its robustness in real-world scenarios. The model is trained for 5000 epochs with a batch size of 2 and patch size of $800 \times 1200$, following (Cui et al., 2023a; Guo et al., 2022).

In addition to daytime scenes, our model is further evaluated in nighttime scenes. Two commonly used datasets are adopted, *i.e.,* GTA5 (Yan et al., 2020) and NHR (Zhang et al., 2020). Given nighttime hazy images, the ground truth images of these two datasets are nighttime clean images and daytime clean images. The models are trained for 300 epochs on the two datasets with a learning rate of $1e^{-4}$.

Since image dehazing plays an essential role in remote sensing, we evaluate our model on a remote sensing dataset, SateHaze1k (Huang et al., 2020), which consists of three sub-sets with different levels of haze degradations: thin, moderate, and thick. The model is separately trained for three datasets for 1000 epochs with a batch size of 32 and a learning rate of $8e^{-4}$.

**Image Defocus Deblurring.** Consistent with recent algorithms (Ruan et al., 2022; Cui et al., 2023a; Zamir et al., 2022a), we use the DPDD (Abuolaim & Brown, 2020) datasets for evaluation. This dataset comprises 350, 74, and 76 scenes for training, validation, and testing. There are four images in each scene, named center view, left view, right view, and an all-in-focus ground truth. Our model is trained under the single-image setting by taking the center-view images as input and computing loss values between the output and ground truth. Our training strategy is identical to that of algorithms (Cui et al., 2023a; Ruan et al., 2022).

**Image Desnowing.** For this task, we use three widely-adopted datasets for training and testing, *i.e.,* CSD (Chen et al., 2021b), SRRS (Chen et al., 2020), and Snow100K (Liu et al., 2018). The preprocessing for these datasets remains identical to previous methods (Chen et al., 2020; Cui et al., 2023a) for fair comparisons. The models are trained for 2000 epochs.

**Image Deraining.** For image deraining, the model is trained on a compound dataset that is mixed based on (Fu et al., 2017; Yang et al., 2017; Zhang et al., 2019; Li et al., 2016), following (Zamir et al., 2022a; Cui et al., 2023b), and tested on the Test2800 (Fu et al., 2017) dataset. The PSNR/SSIM scores are measured using the Y channel in the YCbCr color space, which is consistent with existing methods (Zamir et al., 2022a; Cui et al., 2023b). The model is trained for 300 epochs with a batch size of 4 and a learning rate of $1e^{-4}$.

**Low-Light Image Enhancement.** For this task, the model is evaluated on LOL-v2-synthetic (Yang et al., 2021), which consists of 900 and 100 paired images for training and testing, respectively. The model is trained for 2200 epochs on $3 \times 128 \times 128$ patches.

## B    MORE ABLATION STUDIES

In this section, more ablation results on general image restoration are provided. We first investigate the influence of the number of prompts in general image restoration. Table 21 shows that the performance improves as we increase the number of paired prompts from 1 to 5. However, more prompts lead to degraded performance, which is probably because of overfitting. As a consequence, we finally chose five paired prompts for better results.

Table 21: Ablation studies for the number of paired prompts.

| Number | 1 | 3 | 5 | 7 | 11 |
|---|---|---|---|---|---|
| PSNR | 34.01 | 35.10 | **35.18** | 35.03 | 34.93 |

Equipped with our mechanism, CNN-based and Transformer-based backbones achieve state-of-the-art performance on two kinds of image restoration tasks. It is necessary to compare our backbone, *i.e.,* without using prompting modules, to previous state-of-the-art algorithms. To this end, we conduct experiments on multiple datasets for different tasks using our baseline models and keep the training configurations identical to the final models. From Table 22 to Table 25, we can see that the baseline model is inferior to previous algorithms. When employing our proposed method, they achieve the state-of-the-art performance, demonstrating the effectiveness of our design.

Table 22: Ablation studies on the SOTS (Li et al., 2018) dataset for image dehazing.

| Method | SOTS-Indoor | | SOTS-Outdoor | |
|---|---|---|---|---|
| | PSNR | SSIM | PSNR | SSIM |
| Baseline | 39.07 | 0.995 | 34.98 | 0.993 |
| FocalNet | 40.82 | 0.996 | 37.71 | 0.995 |
| DEA-Net | 40.20 | 0.993 | 36.03 | 0.989 |
| FSNet-S | 40.47 | 0.996 | 37.24 | 0.994 |
| MB-TaylorFormer-B | 40.71 | 0.992 | 37.42 | 0.989 |
| **Ours** | 40.86 | 0.996 | 37.86 | 0.995 |

Table 23: Ablation studies on the Test2800 (Fu et al., 2017) dataset for image deraining.

| Method | Baseline | MPRNet | FSNet | **Ours** |
|---|---|---|---|---|
| PSNR | 33.57 | 33.64 | 33.64 | 33.72 |
| SSIM | 0.936 | 0.938 | 0.936 | 0.937 |

Table 24: Ablation studies on SRRS (Chen et al., 2020) and Snow100K (Liu et al., 2018) for image desnowing.

| Method | SRRS | | Snow100K | |
|---|---|---|---|---|
| | PSNR | SSIM | PSNR | SSIM |
| Baseline | 30.16 | 0.98 | 33.13 | 0.95 |
| MSP-Former | 30.76 | 0.95 | 33.43 | 0.96 |
| FSNet-S | 31.39 | 0.98 | 33.36 | 0.95 |
| FocalNet | 31.34 | 0.98 | 33.53 | 0.95 |
| **Ours** | 31.78 | 0.98 | 33.61 | 0.95 |

## C    COMPLEXITY COMPARISONS

We compare our method with state-of-the-art algorithms on image dehazing and all-in-one image restoration. Figure 6 shows that our model outperforms the recent Transformer-based MB-

Table 25: Ablation studies on the DPDD (Abuolaim & Brown, 2020) dataset for image defocus deblurring.

| Method | Indoor Scenes | | | | Outdoor Scenes | | | | Combined | | | |
|---|---|---|---|---|---|---|---|---|---|---|---|---|
| | PSNR↑ | SSIM↑ | MAE↓ | LPIPS↓ | PSNR | SSIM | MAE | LPIPS | PSNR | SSIM | MAE | LPIPS |
| Baseline | 28.72 | 0.878 | 0.025 | 0.147 | 23.21 | 0.750 | 0.503 | 0.209 | 25.89 | 0.812 | 0.039 | 0.178 |
| Restormer | 28.87 | 0.882 | 0.025 | 0.145 | 23.24 | 0.743 | 0.050 | 0.209 | 25.98 | 0.811 | 0.038 | 0.178 |
| FocalNet | 29.10 | 0.876 | 0.024 | 0.173 | 23.41 | 0.743 | 0.049 | 0.246 | 26.18 | 0.808 | 0.037 | 0.210 |
| Lin *et al.* | 29.11 | 0.889 | - | - | 23.35 | 0.748 | - | - | 26.15 | 0.817 | - | - |
| FSNet | 29.14 | 0.878 | 0.024 | 0.166 | 23.45 | 0.747 | 0.050 | 0.246 | 26.22 | 0.811 | 0.037 | 0.207 |
| **Ours** | 29.38 | 0.883 | 0.023 | 0.145 | 23.49 | 0.753 | 0.049 | 0.208 | 26.35 | 0.816 | 0.036 | 0.178 |

TaylorFormer-B (Qiu et al., 2023) with comparable complexity. Furthermore, as illustrated in Figure 7, our network achieves a significant performance gain over the PromptIR (Li et al., 2024) method in the all-in-one setting, consuming lower complexity. The results demonstrate the efficiency of our design.

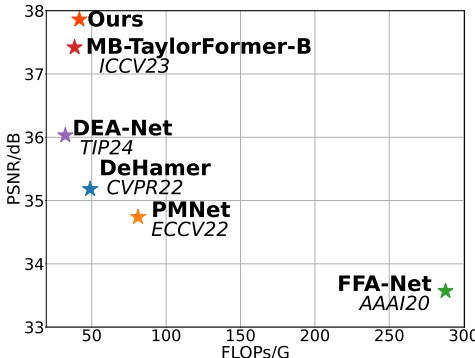

Figure 6: FLOPs *vs.* PSNR on the SOTS-Outdoor (Li et al., 2018) dataset for image dehazing.

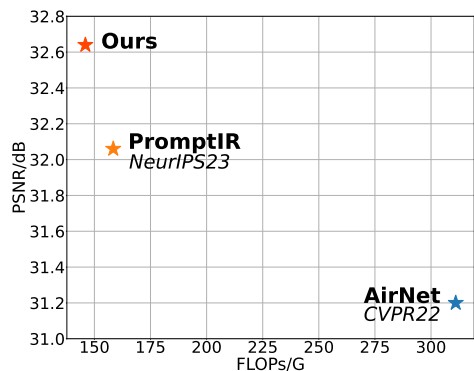

Figure 7: FLOPs *vs.* PSNR for all-in-one models under the three-task setting.

## D    VISUAL COMPARISONS

In this section, we provide visual comparisons for general and all-in-one image restoration tasks.

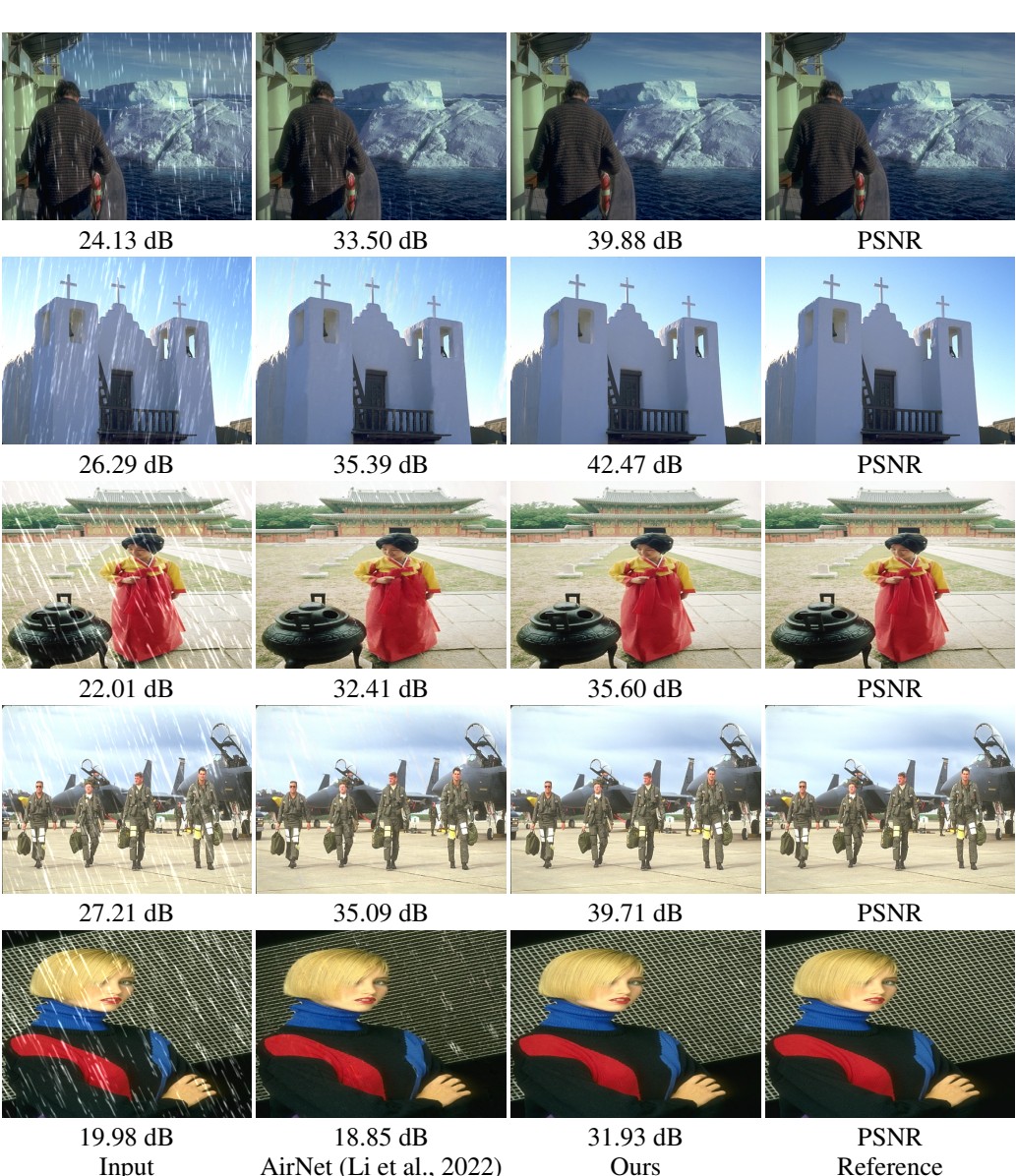

| 24.13 dB | 33.50 dB | 39.88 dB | PSNR |
| 26.29 dB | 35.39 dB | 42.47 dB | PSNR |
| 22.01 dB | 32.41 dB | 35.60 dB | PSNR |
| 27.21 dB | 35.09 dB | 39.71 dB | PSNR |
| 19.98 dB | 18.85 dB | 31.93 dB | PSNR |
| Input | AirNet (Li et al., 2022) | Ours | Reference |

Figure 8: Image deraining comparisons on the Rain100L (Yang et al., 2019) dataset under the single-task setting.

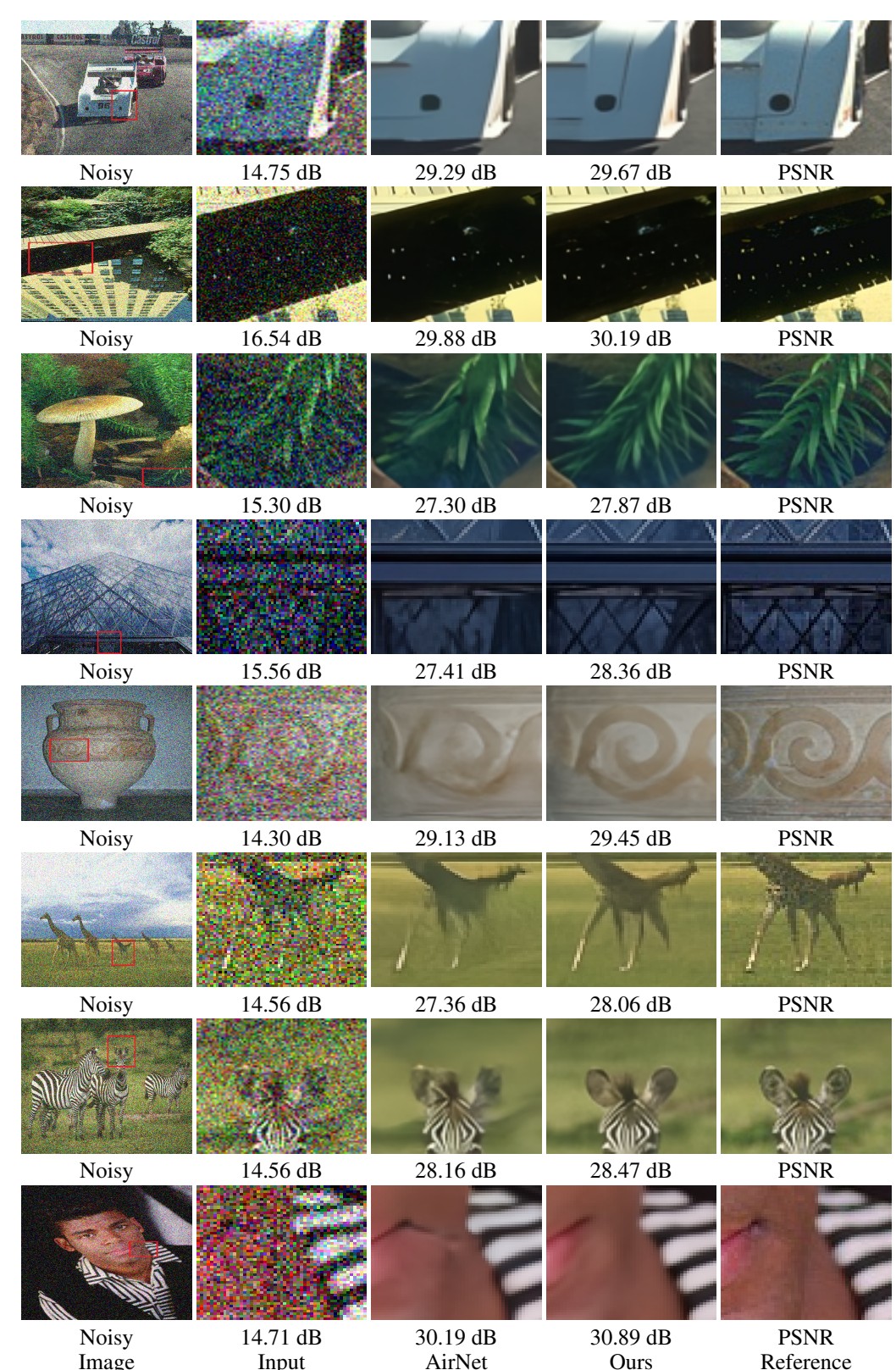

Figure 9: Image denoising comparisons on BSD68 (Martin et al., 2001) with $\sigma = 50$ under the single-task setting.

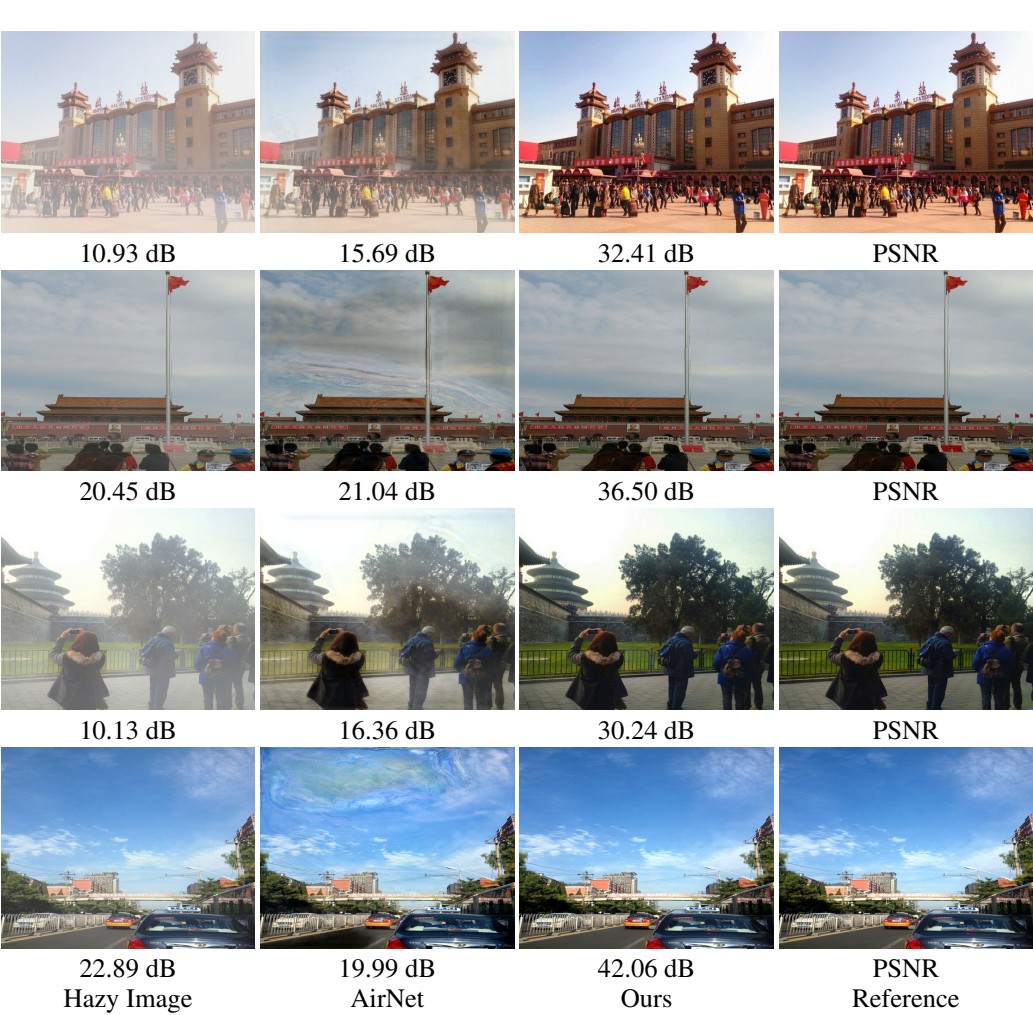

10.93 dB      15.69 dB      32.41 dB      PSNR

20.45 dB      21.04 dB      36.50 dB      PSNR

10.13 dB      16.36 dB      30.24 dB      PSNR

22.89 dB      19.99 dB      42.06 dB      PSNR

Hazy Image      AirNet      Ours      Reference

Figure 10: Image dehazing comparisons on the SOTS-Outdoor (Li et al., 2018) dataset under the single-task setting.

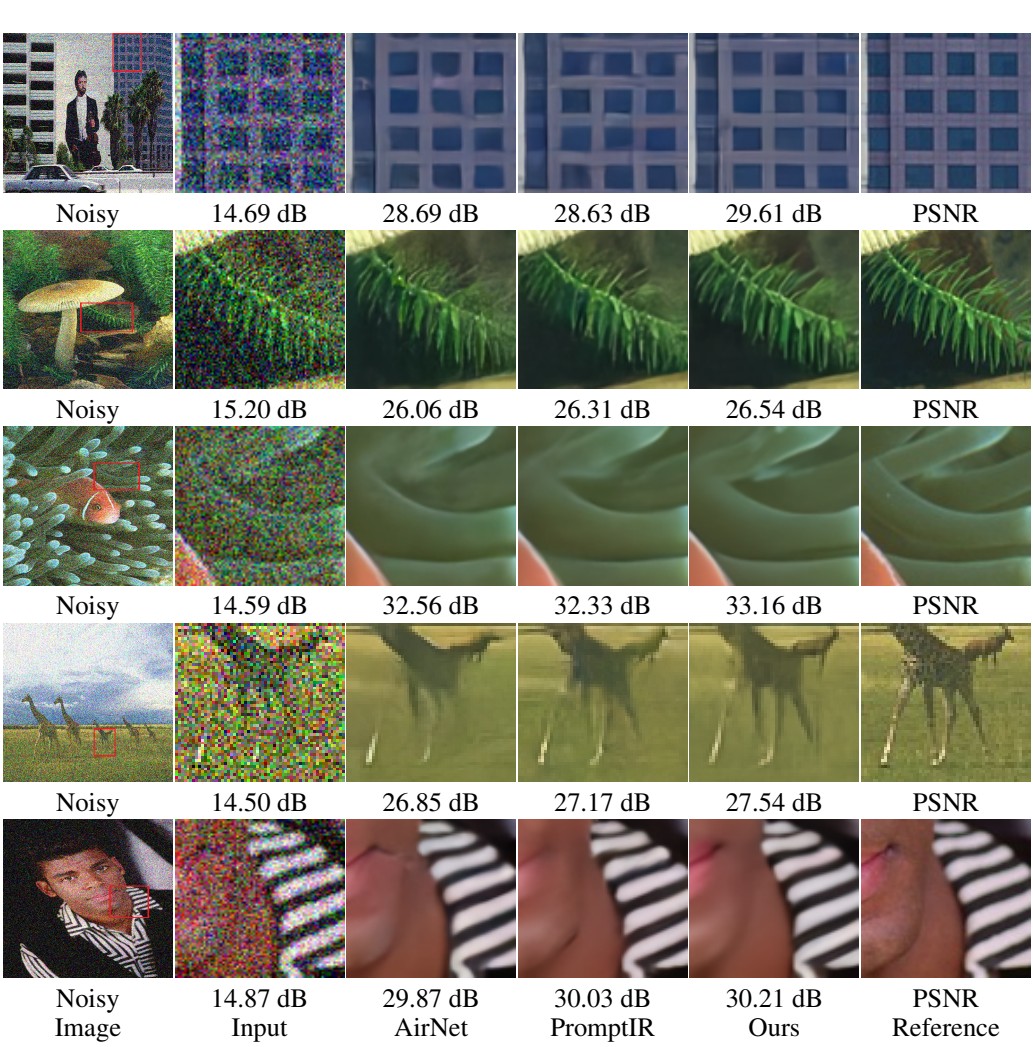

| Noisy | 14.69 dB | 28.69 dB | 28.63 dB | 29.61 dB | PSNR |
| Noisy | 15.20 dB | 26.06 dB | 26.31 dB | 26.54 dB | PSNR |
| Noisy | 14.59 dB | 32.56 dB | 32.33 dB | 33.16 dB | PSNR |
| Noisy | 14.50 dB | 26.85 dB | 27.17 dB | 27.54 dB | PSNR |
| Noisy | 14.87 dB | 29.87 dB | 30.03 dB | 30.21 dB | PSNR |
| Image | Input | AirNet | PromptIR | Ours | Reference |

Figure 11: Image denoising comparisons on BSD68 (Martin et al., 2001) with $\sigma = 50$ under the three-task setting.

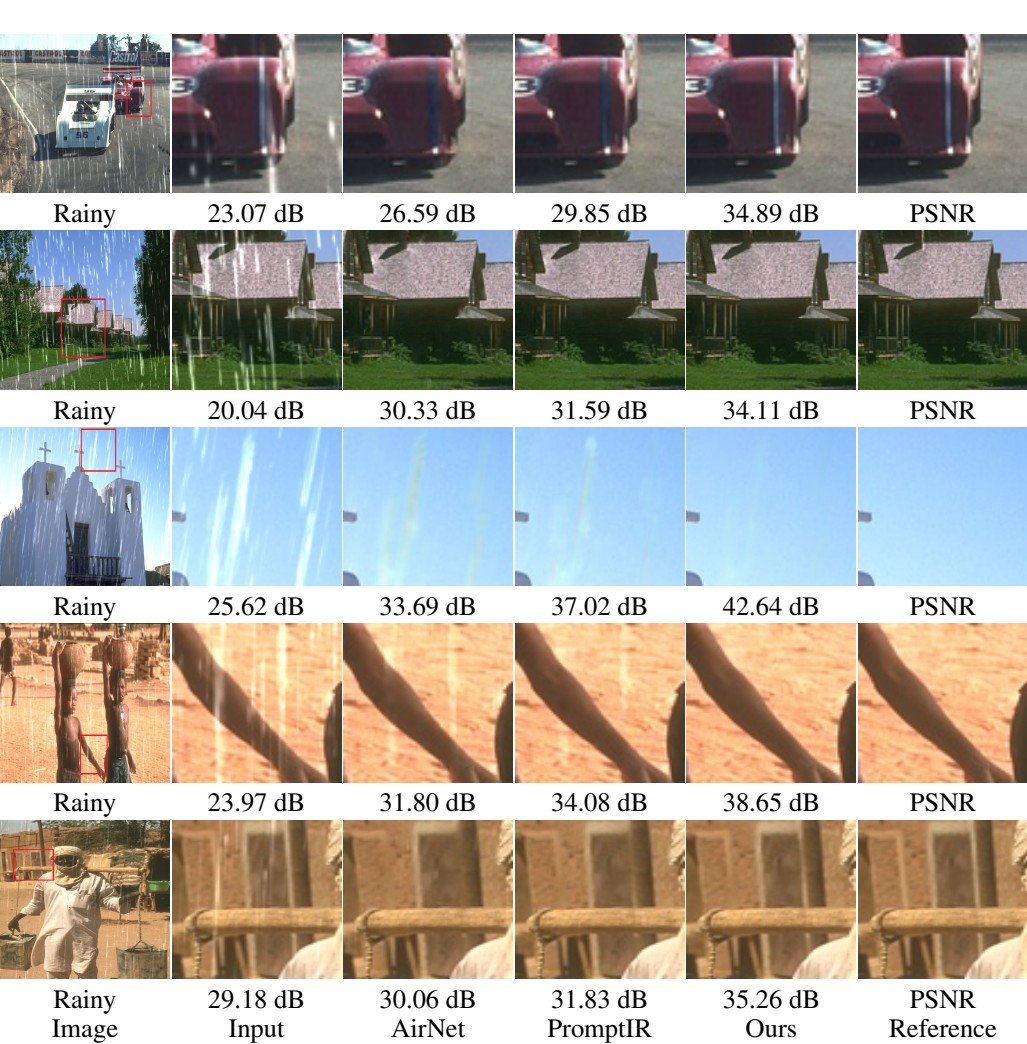

| Rainy | 23.07 dB | 26.59 dB | 29.85 dB | 34.89 dB | PSNR |
| Rainy | 20.04 dB | 30.33 dB | 31.59 dB | 34.11 dB | PSNR |
| Rainy | 25.62 dB | 33.69 dB | 37.02 dB | 42.64 dB | PSNR |
| Rainy | 23.97 dB | 31.80 dB | 34.08 dB | 38.65 dB | PSNR |
| Rainy Image | 29.18 dB Input | 30.06 dB AirNet | 31.83 dB PromptIR | 35.26 dB Ours | PSNR Reference |

Figure 12: Image deraining comparisons on Rain100L (Yang et al., 2019) under the three-task setting.

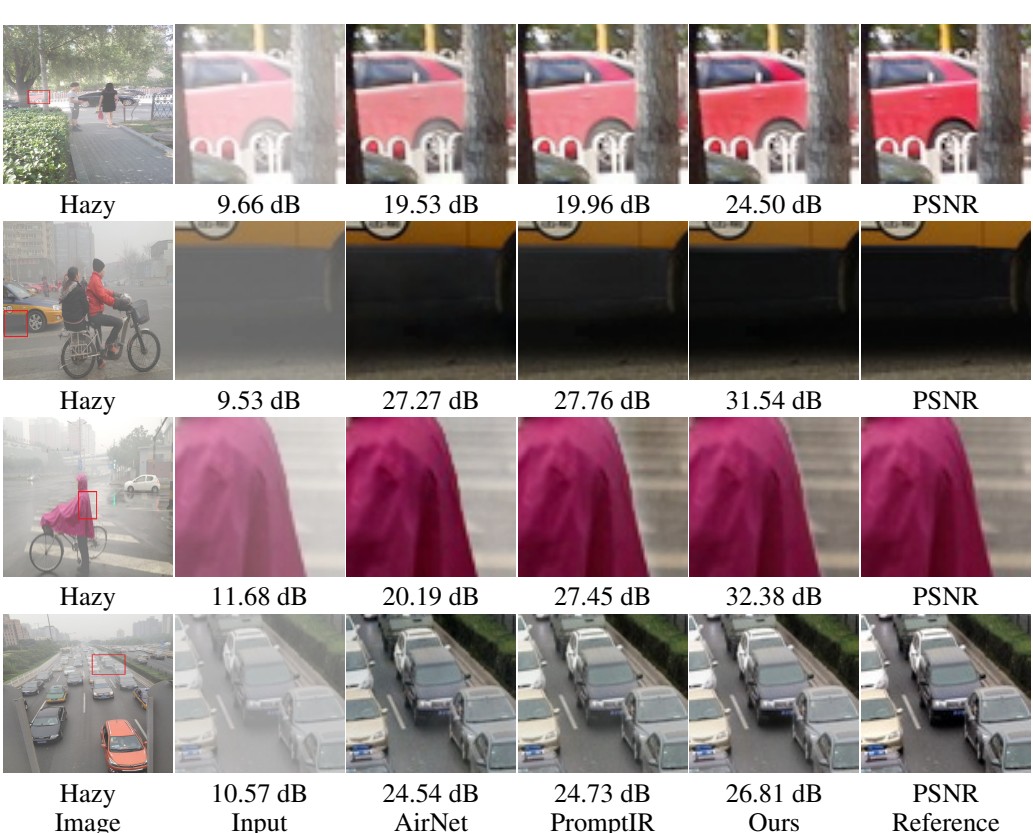

| Hazy | 9.66 dB | 19.53 dB | 19.96 dB | 24.50 dB | PSNR |
| Hazy | 9.53 dB | 27.27 dB | 27.76 dB | 31.54 dB | PSNR |
| Hazy | 11.68 dB | 20.19 dB | 27.45 dB | 32.38 dB | PSNR |
| Hazy Image | 10.57 dB Input | 24.54 dB AirNet | 24.73 dB PromptIR | 26.81 dB Ours | PSNR Reference |

Figure 13: Image dehazing comparisons on SOTS-Outdoor (Li et al., 2018) under the three-task setting.

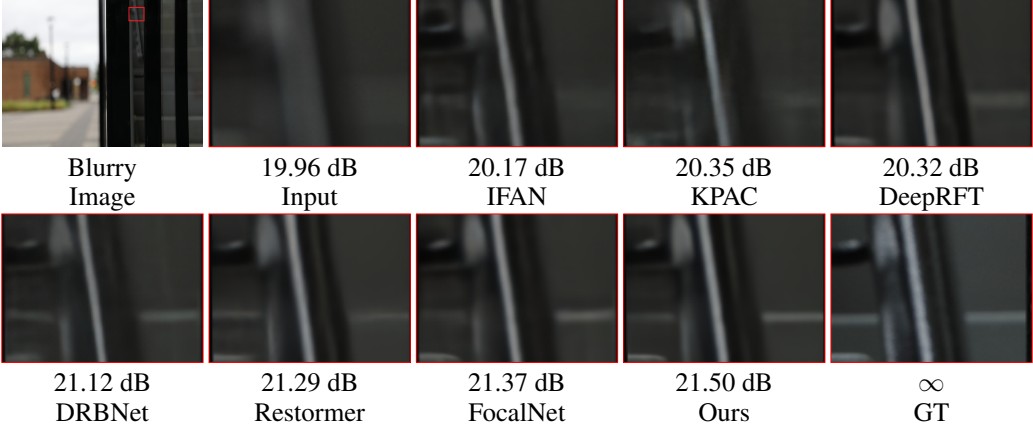

| Blurry Image | 19.96 dB Input | 20.17 dB IFAN | 20.35 dB KPAC | 20.32 dB DeepRFT |
| 21.12 dB DRBNet | 21.29 dB Restormer | 21.37 dB FocalNet | 21.50 dB Ours | ∞ GT |

Figure 14: Image defocus deblurring results on the DPDD (Abuolaim & Brown, 2020) dataset.

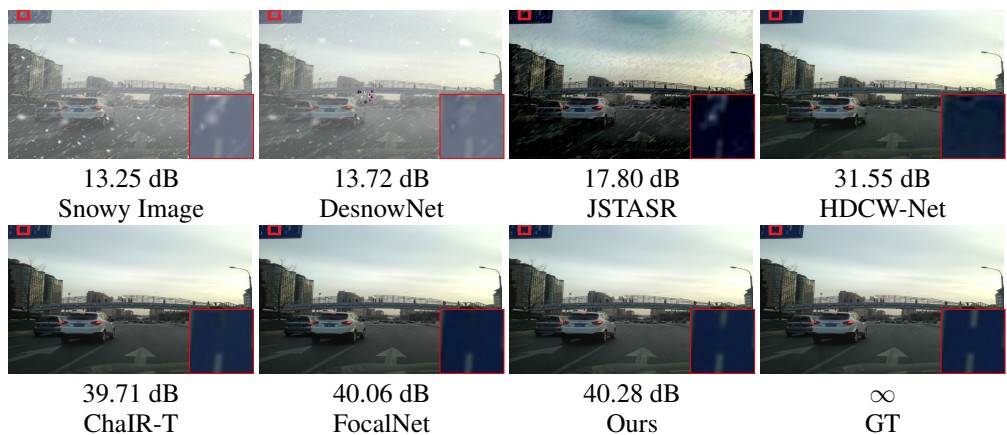



| 13.25 dB | 13.72 dB | 17.80 dB | 31.55 dB |
|:---:|:---:|:---:|:---:|
| Snowy Image | DesnowNet | JSTASR | HDCW-Net |

| 39.71 dB | 40.06 dB | 40.28 dB | ∞ |
|:---:|:---:|:---:|:---:|
| ChaIR-T | FocalNet | Ours | GT |



Figure 15: Image desnowing comparisons on the CSD (Chen et al., 2021b) dataset.

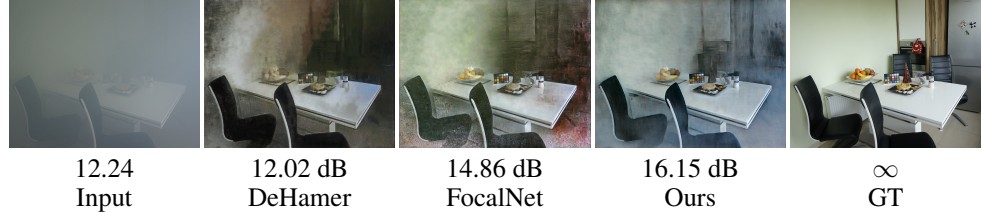



| 12.24 | 12.02 dB | 14.86 dB | 16.15 dB | ∞ |
|:---:|:---:|:---:|:---:|:---:|
| Input | DeHamer | FocalNet | Ours | GT |



Figure 16: Image dehazing comparisons on the real-world Dense-Haze (Ancuti et al., 2019) dataset.

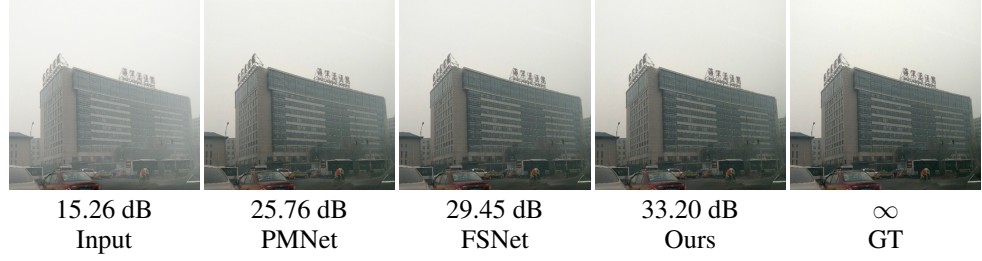



| 15.26 dB | 25.76 dB | 29.45 dB | 33.20 dB | ∞ |
|:---:|:---:|:---:|:---:|:---:|
| Input | PMNet | FSNet | Ours | GT |



Figure 17: Image dehazing comparisons on the Haze4K (Liu et al., 2021) dataset.

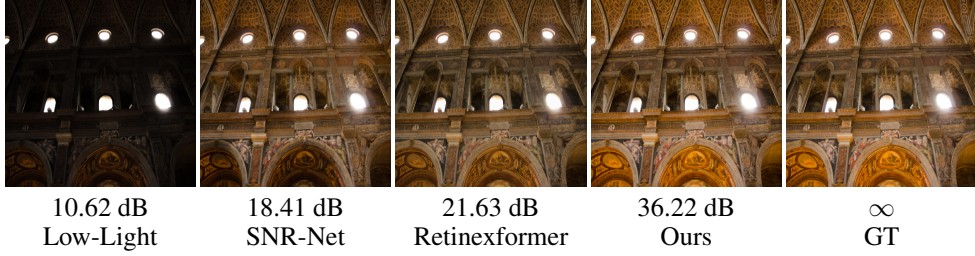



| 10.62 dB | 18.41 dB | 21.63 dB | 36.22 dB | ∞ |
|:---:|:---:|:---:|:---:|:---:|
| Low-Light | SNR-Net | Retinexformer | Ours | GT |



Figure 18: Low-light image enhancement comparisons on the LOL-v2 (Yang et al., 2021) dataset.

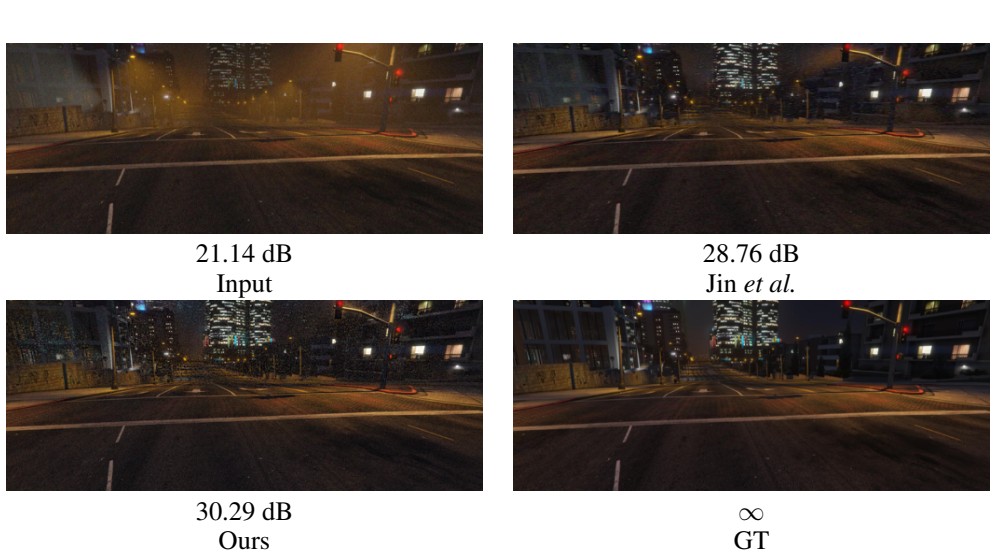

21.14 dB
Input

28.76 dB
Jin *et al.*

30.29 dB
Ours

∞
GT

Figure 19: Image dehazing comparisons on the GTA5 (Yan et al., 2020) dataset.

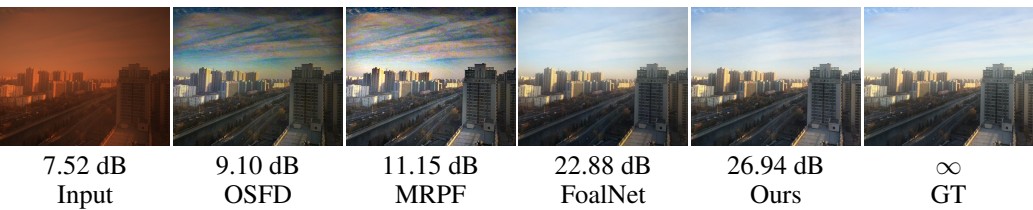

| 7.52 dB | 9.10 dB | 11.15 dB | 22.88 dB | 26.94 dB | ∞ |
| Input | OSFD | MRPF | FoalNet | Ours | GT |

Figure 20: Image dehazing comparisons on the NHR (Zhang et al., 2020) dataset.

