# OpenReview forum: "When Prompt Meets Frequency Learning for Efficient Image Restoration"
_ICLR.cc/2025/Conference — ICLR 2025 Conference Withdrawn Submission_

### Official Review · Reviewer_83cA · 2024-11-02

**Soundness:** 2
**Presentation:** 3
**Contribution:** 2
**Rating:** 3
**Confidence:** 4

**Summary:**

The paper introduces a new frequency-based prompt mechanism to enhance image restoration by focusing on frequency learning, which mainly comprises a prompt generation module and a prompt integration module. Comprehensive experiments are conducted on both general and all-in-one image restoration tasks to demonstrate the effectiveness of the proposed methods

**Strengths:**

1.  Most prompt-based methods primarily operate in the spatial domain, whereas the approach proposed in this paper formulates prompts from the perspective of frequency, presenting an interesting methodology.
2. The plug-and-play frequency prompt mechanism proposed in this paper is applicable to both CNN-based and Transformer-based architectures, demonstrating strong performance across multiple datasets.

**Weaknesses:**

1. Although this method is described as frequency learning in prompts, it only employs global average pooling as a low-pass filter in the prompt generation module and utilizes the fast Fourier transform in the Prompt Integration Module, without delving into an understanding of the frequency domain.
There is a lack of in-depth analysis of why the frequency-domain mechanism is effective.  It lacks the differences and advantages of frequency domain prompt and spatial domain prompt in this paper.
2. The design and naming of the modules show a high similarity to PromptIR [1].
The Prompt Generation Module shares the same name and similar design with a module in PromptIR, differing only by the addition of a subtraction branch considered as the high-frequency prompt branch.
The Prompt Integration Module also has a similar name, and it is questionable whether modulating the frequency domain features with channel and spatial attention is appropriate.
It remains uncertain whether the frequency-domain features after the fast Fourier transform align with the principles of channel and spatial attention, or if direct integration without the fast Fourier transform might be more effective.

[1] Potlapalli V, Zamir S W, Khan S H, et al. Promptir: Prompting for all-in-one image restoration[J]. Advances in Neural Information Processing Systems, 2024, 36.

**Questions:**

1. In Equation 3 of Section 3.1, the calculation of the high-frequency component weight w is inconsistent with Figure 1.
In Figure 1, after removing the low-frequency component from input X to obtain the high-frequency component, a global average pooling (GAP) is applied first, whereas in Equation 3, GAP is omitted, proceeding directly to the subsequent 1×1 convolution.
If implemented as shown in Figure 1, the high-frequency component branch also undergoes a low-pass filter, which would mean that the features no longer correspond to true high-frequency components. This is kind of unreasonable.

2. In Figure 4, two images are produced using only the low-frequency and high-frequency prompts. According to the previous description, the visualized frequency domain signals should represent distinct high- and low-frequency components. However, in reality, the frequency domain distinctions between the two images are not apparent, and both seem concentrated in the low-frequency region.
This makes it difficult to discern the roles of the low-frequency prompt and high-frequency prompt.

3. The feature dimensions within the Prompt Generation Module and Prompt Integration Module require further clarification to more clearly convey the specific operations of each module.

---

### Official Review · Reviewer_v9wk · 2024-11-03

**Soundness:** 3
**Presentation:** 4
**Contribution:** 2
**Rating:** 3
**Confidence:** 5

**Summary:**

This paper proposes a plug-and-play frequency prompt method for efficient image restoration. The mechanism mainly consists of a prompt generation module and a prompt integration module to learn to encode different frequency information. The experimental results demonstrate the effectiveness of the method.

**Strengths:**

1. This paper is well organized and easy to understand.

2. The proposed method is validated on a wide range of image restoration tasks.

3. Qualitative and quantitative experiments are adequate.

**Weaknesses:**

1. The novelty of this work is limited in that both frequency learning [1,2,3,4] and prompt learning [5,6,7,8] or combinations of them [9,10] in image restoration have been extensively explored in past research.

2. The title of the paper emphasizes **efficient** image restoration, however, this paper does not provide a comprehensive and thorough comparison of efficiency. In addition to the Part C results in the Appendix, a full comparison of parameters, FLOPs, and on-device inference latency is recommended.

3. The training setup is very unfair. The vast majority of algorithms are trained using only L1 loss. Additional performance gains may stem from frequency losses. In addition, the method uses a much larger patch size 256 $\times$ 256. This significantly weakens the persuasiveness of the main experimental results of this paper.

> 1. Selective Frequency Network for Image Restoration. ICLR 2023

> 2. Focal Network for Image Restoration. ICCV 2023

> 3. IRNeXt: Rethinking Convolutional Network Design for Image Restoration. ICML 2023

> 4. Hybrid Frequency Modulation Network for Image Restoration. IJCAI 2024

> 5. PromptIR: Prompting for All-in-One Blind Image Restoration. NIPS 2023

> 6. Prompt-In-Prompt Learning for Universal Image Restoration. arXiv preprint arXiv:2312.05038.

> 7. Towards Effective Multiple-in-One Image Restoration: A Sequential and Prompt Learning Strategy, arXiv preprint arXiv:2401.03379.

> 8. Multimodal Prompt Perceiver: Empower Adaptiveness Generalizability and Fidelity for All-in-One Image Restoration.CVPR 2024

> 9. AdaIR: Adaptive All-in-One Image Restoration via Frequency Mining and Modulation. arXiv preprint arXiv:2403.14614.

> 10. Seeing the Unseen: A Frequency Prompt Guided Transformer for Image Restoration. ECCV 2024

**Questions:**

Please see weaknesses.

---

### Official Review · Reviewer_Azqa · 2024-11-04

**Soundness:** 2
**Presentation:** 2
**Contribution:** 2
**Rating:** 5
**Confidence:** 5

**Summary:**

This work utilizes the frequency information to prompt image restoration, achieving good performance across multiple specific and all-in-one tasks. Despite the simplicity of the proposed method, the authors conducted extensive experiments to demonstrate its effectiveness.

**Strengths:**

1.The authors performed numerous experiments to verify the effectiveness of the method.
2.The proposed method can be used as a plug-and-play approach to enhance the performance of existing models.

**Weaknesses:**

1.The comparison methods do not seem to include recent well-performing works, raising doubts about the competitiveness of the experimental results, given only minor performance improvements.
2.In the abstract, the authors stated, “However, these algorithms mostly operate in the spatial domain.” However, from Figure 1, it appears that the proposed method also operates in the spatial domain, even though it influences spectral changes.
3.In related work, the authors mentioned, “Prompt learning is originally used in natural language processing to fine-tune a trained model to a downstream task by providing in-context and task-specific information.” However, it seems the proposed method in the all-in-one task can only provide a general prompt, rather than a specific prompt, which differs from the authors' expectations of prompt learning.

**Questions:**

1.I am curious why, given the emphasis on the proposed method’s effectiveness in specific tasks, the authors did not compare it to other all-in-one restoration models in specific tasks.
2.The authors used global average pooling to obtain the so-called low-frequency component while emphasizing the high-frequency component in another branch. But why use global average pooling again in the high-frequency branch? Is this emphasizing the low-frequency part of the high-frequency component? I am curious about the rationale behind this design, as the authors did not seem to provide a clear explanation.

---

### Note · Authors · 2024-11-12

I have read and agree with the venue's withdrawal policy on behalf of myself and my co-authors.